# Evaluating Parameter Efficient Methods for RLVR

**Qingyu Yin** [* 1]   **Yulun Wu** [* 1]   **Zhennan Shen** [* 2]   **Sunbowen Li** [3]   **Zhilin Wang** [4]
**Yanshu Li** [5]   **Chak Tou Leong** [6]   **Jiale Kang** [7]   **Jinjin Gu** [8]

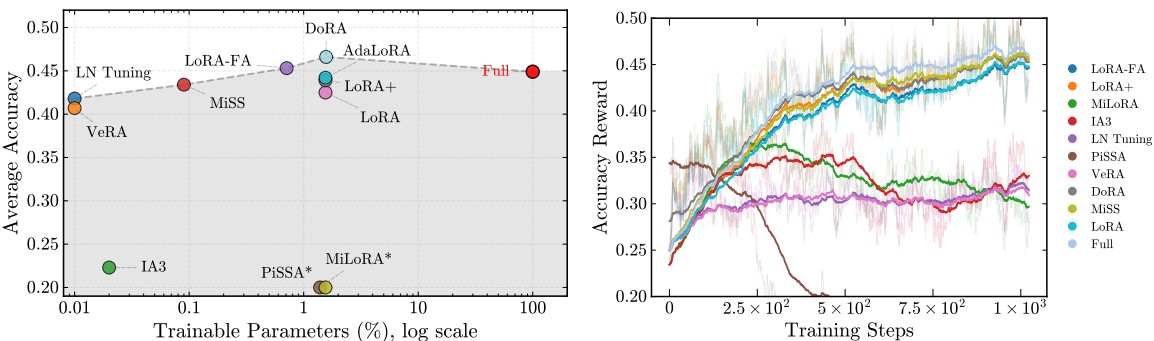

*Figure 1.* **Left:** Comparison of average accuracy *vs.* percentage of trainable parameters (log scale) for various parameter efficient methods under our RLVR evaluations. The shaded area represents the performance frontier. **Right:** Training dynamics showing accuracy reward over training steps for different methods.

## Abstract

We systematically evaluate Parameter-Efficient Fine-Tuning (PEFT) methods under the paradigm of Reinforcement Learning with Verifiable Rewards (RLVR). RLVR incentivizes language models to enhance their reasoning capabilities through verifiable feedback; however, while methods like LoRA are commonly used, the optimal PEFT architecture for RLVR remains unidentified. In this work, we conduct the first comprehensive evaluation of over 12 PEFT methodologies across the DeepSeek-R1-Distill families on mathematical reasoning benchmarks. Our empirical results challenge the default adoption of standard LoRA with three main findings. First, we demonstrate that structural variants, such as DoRA, AdaLoRA, and MiSS, consistently outperform LoRA. Second, we uncover a spectral collapse phenomenon in SVD-informed initialization strategies (*e.g.,* PiSSA, MiLoRA), attributing their failure to a fundamental misalignment between principal-component updates and RL optimization. Fur-

thermore, our ablations reveal that extreme parameter reduction (*e.g.,* VeRA, Rank-1) severely bottlenecks reasoning capacity. We further conduct ablation studies and scaling experiments to validate our findings. This work provides a definitive guide for advocating for more exploration for parameter-efficient RL methods.

## 1. Introduction

Large Language Models (LLMs) (Vaswani et al., 2017; Brown et al., 2020) have demonstrated remarkable proficiency in complex reasoning tasks, particularly within mathematical and scientific domains. Recently, Reinforcement Learning with Verifiable Rewards (RLVR) (Guo et al., 2025; Yu et al., 2025) has emerged as the dominant paradigm for further amplifying these reasoning capabilities, enabling models to transcend the limitations of supervised fine-tuning.

Despite these capabilities, the training process of RL remains notoriously complex and resource-intensive (Ouyang et al., 2022), necessitating the development of more efficient training methods. A key distinction contributing to this inefficiency is the nature of supervision: unlike Supervised Fine-Tuning (SFT), which benefits from dense knowledge transfer via teacher-forcing, RL (specifically RLVR) relies on sparse supervision, typically manifesting as a 1-bit reward signal (Uesato et al., 2022). Mechanistically, this sparsity leads to updates being confined to small subnets (Frankle & Carbin, 2018; Mukherjee et al., 2025)

[1]Zhejiang University [2]HKUST [3]WUST [4]USTC [5]Brown University [6]Hong Kong Polytechnic University [7]Independent [8]INSAIT, Sofia University "St. Kliment Ohridski". Correspondence to: Qingyu Yin <qingyu.yin@zju.edu.cn>.

*Proceedings of the 43rd International Conference on Machine Learning*, Seoul, South Korea. PMLR 306, 2026. Copyright 2026 by the author(s).

or sparse parameters (Zhu et al., 2025), implying significant parameter redundancy during full-parameter RL training. Consequently, there is substantial scope for optimizing RL through parameter-efficient approaches. Recent works (Wang et al., 2025b) have demonstrated that Low-Rank Adaptation (LoRA) (Hu et al.)—which decomposes weight updates into low-rank matrices to reduce computational cost—can yield competitive performance compared to full-parameter training.

**Research Question.** While a proliferation of LoRA variants and PEFT methods has emerged, the application of these techniques in reinforcement learning remains predominantly confined to standard LoRA. This predominance raises a critical uncertainty regarding whether the standard LoRA architecture truly represents the optimal strategy for the distinct optimization dynamics of RL, considering there are many other PEFT variants *e.g.,* DoRA (Liu et al., 2024) that have been verified to be stronger than LoRA under the fine-tuning scenarios. This anchors our primary research question:

> **Which Parameter-Efficient method is best suited for Reinforcement Learning?**

To address this, we conduct the first large-scale, comprehensive evaluation of PEFT methods within RL. Through a multidimensional analysis, we derive actionable insights to guide the community in navigating the development of Parameter-Efficient Reinforcement Learning.

**Experimental settings.** To rigorously investigate these dynamics, we construct a large-scale benchmark using the `DeepSeek-R1-Distill` (DeepSeek-AI, 2025) model families. Our experiments span mathematical reasoning tasks including MATH-500 (Lightman et al., 2023), AIME (Zhang & Math-AI, 2024; 2025), AMC (Li et al., 2024), *etc.,* utilizing the RLVR framework on TRL (von Werra et al., 2020). We evaluate over 12 PEFT variants categorized into structural, initialization-based, and efficiency-driven methods. All methods are tested under controlled conditions with unified hyperparameters (*e.g.,* learning rate, batch size, and rank) to ensure a fair comparison across the distinct optimization landscapes of each adapter.

**Key Findings.** Our large-scale empirical analysis challenges the default adoption of standard LoRA, highlighted by three core insights: (1) **Structural variants surpass standard LoRA.** We find that standard LoRA is suboptimal for RLVR. Structural variants *e.g.,* DoRA (Liu et al., 2024), MiSS (Kang & Yin, 2025), and AdaLoRA (Zhang et al., 2023b) consistently yield superior reasoning accuracy, with DoRA notably outperforming even full-parameter fine-tuning. (2) **SVD-based initialization suffers from spectral misalignment.** Strategies prioritizing principal components *e.g.,* PiSSA (Meng et al., 2024) experience training col-

lapse, which we attribute to a fundamental conflict with RL's intrinsic *off-principal* update dynamics. Conversely, initialization methods based on learning rate adjustment *e.g.,* LoRA+ (Hayou et al., 2024) prove highly robust. (3) **Less is not always more for Parameter-Efficient RLVR.** Our findings reveals that, while RLVR can tolerate moderate parameter reduction, *e.g.,* freezing half of the weights in LoRA-FA, it exhibits a strict lower bound on expressivity. Extreme compression schemes, such as VeRA (Kopiczko et al., 2023), Rank-1 adapters, or exclusive LayerNorm tuning, act as a structural bottleneck that causes performance to collapse, failing to sustain the acquisition of complex reasoning behaviors.

**Ablation and Scaling.** Extensive ablations across batch sizes, ranks, and learning rates further substantiate the robustness of our conclusions. Crucially, we extend our validation by scaling to the larger DeepSeek-R1-Distill-Qwen-7B model. The consistent superiority of structural variants across these varying parameter scales confirms that our insights are intrinsic to the RLVR optimization landscape and hold firm regardless of model capacity.

**Contributions.** To the best of our knowledge, this work represents the first systematic study bridging the gap between diverse PEFT methodologies and the specific optimization dynamics of Reinforcement Learning with Verifiable Rewards. Our contributions are summarized as follows:

- **First Comprehensive PEFT-RLVR Benchmark** (Section 2). We establish a large-scale benchmark evaluating over 12 parameter-efficient methods. We demonstrate that the prevailing practice of defaulting to standard LoRA is suboptimal for RLVR.

- **Superiority of Structural Variants** (Section 3.1). We empirically demonstrate that structural variants consistently outperform standard LoRA and frequently surpass full-parameter fine-tuning.

- **Mechanism of SVD-based Initialization Failure** (Section 3.1). We uncover a critical failure mode in SVD-informed initialization strategies. Through spectral analysis, we provide a mechanistic explanation: these methods enforce updates on principal components, creating a fundamental structural misalignment with RLVR's intrinsic tendency to operate in the off-principal regime.

- **Identification of the Expressivity Floor** (Section 3.1). We identify a distinct performance boundary in parameter efficiency. Our results reveal that extreme parameter reduction methods create an information bottleneck that severely limits the plasticity required for reasoning.

- **Scalability and Robustness** (Section 3.2 and 3.3). We validate the generalizability of our findings by scaling experiments to the 7B parameter regime and conducting extensive ablations on batch sizes, learning rates, and ranks.

## 2. Preliminaries & Setup

### 2.1. Reinforcement Learning with Verifiable Rewards

**Group Relative Policy Optimization (GRPO).** Reinforcement Learning with Verifiable Rewards (RLVR) has emerged as a powerful paradigm for enhancing LLM reasoning by utilizing deterministic verifiers to provide sparse but accurate binary rewards (Yu et al., 2025; Shao et al., 2024). Unlike traditional RLHF, RLVR leverages rule-based feedback (e.g., math correctness or code execution) to elicit complex behaviors such as self-correction and iterative refinement (Shao et al., 2024; Liu et al., 2025). The foundational framework for many recent advancements is Group Relative Policy Optimization (GRPO), which eliminates the need for a separate critic model by estimating advantages through group statistics (Shao et al., 2024). For a given prompt $q$, GRPO samples a group of $G$ responses $\{o_1, \ldots, o_G\}$ and optimizes the following surrogate objective:

$$\mathcal{J}_{\text{GRPO}}(\theta) = \mathbb{E}_{q \sim \mathcal{D}, \{o_i\} \sim \pi_{\theta_{\text{old}}}} \left[ \frac{1}{G} \sum_{i=1}^{G} \frac{1}{|o_i|} \sum_{t=1}^{|o_i|} \right.$$
$$\left. \min \left( r_{i,t}(\theta) \hat{A}_i, \text{clip} \left( r_{i,t}(\theta), 1 \pm \epsilon \right) \hat{A}_i \right) \right] \quad (1)$$

where $\hat{A}_i = \frac{(R_i - \text{mean}(\{R_j\}))}{\text{std}(\{R_j\})}$ represents the standardized advantage within the group (Shao et al., 2024).

**GRPO Variants and Improvements.** To address challenges such as entropy collapse and training instability in long CoT scenarios, several optimized variants have been proposed. Decoupled Clip and Dynamic sampling Policy Optimization (DAPO) introduces a *Clip-Higher* strategy, which decouples the clipping range into $\epsilon_{low}$ and $\epsilon_{high}$ (Yu et al., 2025). By setting a larger $\epsilon_{high}$ *e.g.,* 0.28, DAPO provides more room for low-probability exploration tokens to be uplifted, effectively maintaining policy diversity (Yu et al., 2025). Furthermore, DAPO employs *Dynamic Sampling* to filter out prompts where all outputs yield identical rewards *e.g.,* all 0 or all 1, ensuring consistent gradient signals and improved sample efficiency (Yu et al., 2025). Another significant refinement is Dr. GRPO, which identifies and mitigates systematic biases inherent in the original GRPO formulation (Liu et al., 2025). Dr. GRPO removes the per-response length normalization term $\frac{1}{|o_i|}$, which inadvertently rewards longer incorrect responses while penalizing concise correct ones. Additionally, Dr. GRPO eliminates the group-level standard deviation in advantage estimation to avoid difficulty bias, where questions with low reward variance (too easy or too hard) receive disproportionately high weights (Liu et al., 2025).

Following the previous works (Shao et al., 2024; Yu et al., 2025; He et al., 2025; Wang et al., 2025b), we adopt **DAPO** as our standard training algorithm and leave other methods as ablation experiments.

### 2.2. PEFT Methods

**Low-Rank Adaptation (LoRA).** LoRA (Hu et al.) hypothesizes that the change in weights during adaptation has a low intrinsic rank. Given a pre-trained weight matrix $\mathbf{W}_0 \in \mathbb{R}^{d_{\text{out}} \times d_{in}}$, LoRA freezes $\mathbf{W}_0$ and constrains the update $\Delta \mathbf{W}$ by decomposing it into the product of two low-rank matrices $\mathbf{B} \in \mathbb{R}^{d_{\text{out}} \times r}$ and $\mathbf{A} \in \mathbb{R}^{r \times d_{in}}$, where the rank $r \ll \min(d_{in}, d_{out})$. The forward pass is formalized as:

$$\mathbf{h} = \mathbf{W}_0 \mathbf{x} + \Delta \mathbf{W} \mathbf{x} = \mathbf{W}_0 \mathbf{x} + \frac{\alpha}{r} \mathbf{B} \mathbf{A} \mathbf{x}, \quad (2)$$

where $\alpha$ is a constant scaling factor. In the standard implementation, $\mathbf{A}$ is initialized with random Gaussian noise, while $\mathbf{B}$ is initialized to zero, ensuring that $\Delta \mathbf{W} = 0$ at the beginning of training.

**PEFT Methods Selection.** While LoRA remains the most prominent approach, the landscape of parameter-efficient methods has expanded significantly to encompass a diverse array of alternative methodologies. To provide a thorough evaluation, we categorize these adopted PEFT methods into five distinct groups based on their design paradigms, as illustrated in Table 1:

- **Baselines**: We employ Full-Parameter Fine-Tuning and standard LoRA (Hu et al.) as our primary benchmarks to establish the performance upper bound and the standard efficiency baseline ($y = \mathbf{W}_0 x + \mathbf{B} \mathbf{A} x$), respectively.

- **Structural Variants**: This category encompasses methods that *fundamentally alter the architectural formulation* beyond the standard additive product $\mathbf{B} \mathbf{A}$. Instead of the fixed low-rank decomposition, these methods introduce novel structural components. We include **DoRA** (Liu et al., 2024), which decouples magnitude and direction ($y = m \frac{\mathbf{W} x}{||\mathbf{W}||}$); **AdaLoRA** (Zhang et al., 2023b), which employs an SVD-like adaptive rank structure ($y = \mathbf{W}_0 x + \mathbf{P} \mathbf{\Lambda} \mathbf{Q} x$); and others like **MiSS** (Kang & Yin, 2025) that utilize distinct sub-network selection.

- **Initialization Strategies**[1]: These methods *retain the standard adapter architecture* but intervene in the initialization state or optimization dynamics to accelerate convergence. We evaluate signal-informed strategies like **PiSSA** (Meng et al., 2024) and **MiLoRA** (Wang et al., 2025a), which initialize matrices $\mathbf{A}$ and $\mathbf{B}$ using the Principal Components (SVD) of $\mathbf{W}_0$ rather than random Gaussian noise. We also examine methods that adjust the training dynamics, such as **LoRA+** (Hayou et al., 2024), which uses differentiated learning rates ($\eta_B \gg \eta_A$), and

---

[1]In this context, we define initialization-based methods broadly to include all strategies that are configured at the onset of training without altering the standard LoRA architecture ($\mathbf{W}_0 + \mathbf{B} \mathbf{A}$), encompassing both weight initialization and the pre-specification of optimization dynamics.

*Table 1.* A variety of PEFT methods are listed, each with its specific update formulation and initialization strategy. LN denotes Layernorm.

| Method | Forward | Initialization |
|---|---|---|
| ***Baseline*** | | |
| Full-Param Fine-Tuning | $y = \boldsymbol{W}_0 x$ | N/A |
| LoRA (Hu et al.) | $y = \boldsymbol{W}_0 x + \frac{\alpha}{r} \boldsymbol{B}\boldsymbol{A}x$ | $\boldsymbol{A} \sim N(0, \sigma^2), \boldsymbol{B} \sim 0$ |
| ***Structural*** | | |
| DoRA (Liu et al., 2024) | $y = \boldsymbol{m}(\ \boldsymbol{W}_0 x + \boldsymbol{B}\boldsymbol{A}x\ /\ \|\boldsymbol{W}_0 + \boldsymbol{B}\boldsymbol{A}\|_c)$ | $\boldsymbol{A} \sim \text{Rect.KaimingUnif}, \boldsymbol{B} \sim 0$ |
| MiSS (Kang & Yin, 2025) | $y = \boldsymbol{W_0} x + \text{expand}(\boldsymbol{D})x$ | $\boldsymbol{D} \sim 0$ |
| AdaLoRA (Zhang et al., 2023b) | $y = \boldsymbol{W}_0 x + \boldsymbol{P}\boldsymbol{\Lambda}\boldsymbol{Q}x$ | $\boldsymbol{\Lambda} \sim 0, \boldsymbol{P}, \boldsymbol{Q} \sim N(0, \sigma^2)$ |
| ***Initialization*** | | |
| PiSSA (Meng et al., 2024) | $y = (\boldsymbol{W}_0 - \boldsymbol{B}\boldsymbol{A})x + \boldsymbol{B}\boldsymbol{A}x$ | $\boldsymbol{A} = U_{[:,:r]}S_{[:r,:r]}^{1/2}, \boldsymbol{B} = S_{[:r,:r]}^{1/2}V_{[:,:r]}^{\top}$ |
| MiLoRA (Wang et al., 2025a) | $y = (\boldsymbol{W}_0 - \boldsymbol{B}\boldsymbol{A})x + \boldsymbol{B}\boldsymbol{A}x$ | $\boldsymbol{A} = U_{[:,r:]}S_{[r:,r:]}^{1/2}, \boldsymbol{B} = S_{[r:,r:]}^{1/2}V_{[:,r:]}^{\top}$ |
| LoRA+ (Hayou et al., 2024) | $y = \boldsymbol{W}_0 x + \boldsymbol{B}\boldsymbol{A}x$ | $\eta_B = \lambda\eta_A$ (Ratio of Learning Rates) |
| rsLoRA (Kalajdzievski, 2023) | $y = \boldsymbol{W}_0 x + \frac{\alpha}{\sqrt{r}}\boldsymbol{B}\boldsymbol{A}x$ | $\boldsymbol{A} \sim N(0, \sigma^2), \boldsymbol{B} \sim 0$ |
| ***Efficiency*** | | |
| LoRA-FA (Zhang et al., 2023a) | $y = \boldsymbol{W}_0 x + \boldsymbol{B}\boldsymbol{A}x$ | $\boldsymbol{A} \sim N(0, \sigma^2)$ (Frozen), $\boldsymbol{B} \sim 0$ |
| VeRA (Kopiczko et al., 2023) | $y = \boldsymbol{W}_0 x + \Lambda_b \boldsymbol{B} \Lambda_d \boldsymbol{A}x$ | $\boldsymbol{A}, \boldsymbol{B}$ Frozen Random, $\boldsymbol{d} \sim 0.1, \boldsymbol{b} \sim 0$ |
| ***Other PEFTs*** | | |
| LN Tuning (Qi et al., 2022) | $y = \frac{g}{\sigma} \odot (x - \mu) + b$ | Pre-trained $g$ (gain) and $b$ (bias) |
| IA$^3$ (Liu et al., 2022a) | $x' = x \odot l$ | $l \sim 1$ (Rescaling vectors for $K, V$, FFN) |

**rsLoRA** (Kalajdzievski, 2023), which employs stable rank scaling factors.

- **Efficiency-Oriented Variants**: Driven by hardware constraints, this category investigates methods designed to minimize memory footprints. We evaluate **LoRA-FA** (Zhang et al., 2023a) (freezing $\boldsymbol{A}$) and **VeRA** (Kopiczko et al., 2023) (freezing random projection matrices and training only scaling vectors).

- **Other PEFT Mechanisms**: Finally, we assess approaches that diverge from the weight-update paradigm entirely. This includes **IA$^3$** (Liu et al., 2022b), which scales activation vectors via element-wise multiplication, and **LayerNorm Tuning** (Qi et al., 2022), to evaluate the efficacy of alternative adaptation mechanisms.

**PEFT Settings.** We benchmark standard LoRA and other PEFT methods. It is worth noting that following Schulman & Lab (2025) and Wang et al. (2025b), we target all linear modules ($\{q, k, v, o, gate, up, down\}\_proj$), as this configuration has been demonstrated to yield superior performance in prior studies (Schulman & Lab, 2025). We set rank 32, dropout rate 0.05, and alpha 64 for all PEFT methods.

### 2.3. Models and Datasets

**Base Models.** Our selection of base models is guided by three key criteria: (1) Following the standard RL paradigm, we select models that have undergone SFT as a cold start phase to ensure sufficient initial reasoning capabilities and reasoning format; (2) We employ models across different parameter scales to disentangle size-specific effects. Based on these principles, we utilize two reasoning models: *DeepSeek-R1-Distill-Qwen-1.5B* (DeepSeek-AI, 2025), and *DeepSeek-R1-Distill-Qwen-7B* (DeepSeek-AI, 2025).

**Training Dataset.** We utilize the *open-r1/DAPO-Math-17k-Processed* dataset (Yu et al., 2025), which comprises approximately 17.4k high-quality mathematical queries and has been validated by prior research. To enforce structured reasoning, we impose a strict output format, requiring the model to enclose reasoning traces within `<think>...</think>` tags and encapsulate the final answer using `\\boxed{}`.

### 2.4. Training Settings

**Reward Mechanism.** We employ a strict outcome-based reward. The generated answers are extracted and verified against ground truth using a combination of `latex2sympy` (Tali, 2016) and `math_verify` (Kydlíček et al., 2026) . The reward is binary: $R = 1$ for mathematically equivalent answers and $R = 0$ otherwise. The overall reward recipe follows the principles of JustRL (He et al., 2025).

**Hyperparameter and Other Details.** We utilize `Accelerate` with DeepSpeed ZeRO-2 optimization (offloading optimizer states) to minimize memory usage. For rollout generation, we employ the `vLLM` engine in co-location mode to maximize throughput. Following the previous settings of LoRA RL training (Wang et al., 2025b), we generate $G = 8$ rollouts per prompt and use a constant learning rate of 1e-5 with no warmup. The training is conducted with a maximum prompt length of 512 and a completion length of 16384 tokens. For the RLVR objective, we set the DAPO epsilon to $0.28$ *i.e., clip-higher* and do not employ a KL coefficient ($\beta$). Regarding the batch configurations, the 1.5B model is trained with a per-device batch size of 4 and a global batch size of 128 over 1,024 steps; for the 7B model, we use a per-device batch size of 1 and a global batch size of 32 over 8,192 steps. In both settings, the gradient accumulation steps are fixed at 8. We do not apply complicated strategies

*e.g.,* multi-stage training, as it has been proven that simple RL scaling (He et al., 2025) can still achieve competitive results.

## 2.5. Evaluation

*Table 2.* Overview of the mathematical reasoning datasets used in our study, including the number of test samples and the specific evaluation metrics (*e.g.,* $Avg@k$) employed for each benchmark.

| Benchmark | Nums | Eval Method |
|---|---|---|
| AIME24 (Zhang & Math-AI, 2025) | 30 | Avg@32 ,Pass@32 |
| AIME25 (Zhang & Math-AI, 2025) | 30 | Avg@32 ,Pass@32 |
| MATH500 (Lightman et al., 2023) | 500 | Avg@4, Pass@4 |
| Minerva (Lewkowycz et al., 2022) | 272 | Avg@4, Pass@4 |
| AMC (Li et al., 2024) | 40 | Avg@32, Pass@32 |
| HMMT (Balunović et al., 2025) | 30 | Avg@32, Pass@32 |

**Benchmark Selections.** We evaluate models using the mathematics suite. following the benchmark selection in Table 2 from previous work including He et al. (2025), Guo et al. (2025) and Wang et al. (2025b). The benchmarks include MATH-500 (Lightman et al., 2023), AMC23 (Li et al., 2024), AIME24/25 (Zhang & Math-AI, 2024; 2025), Minerva (Lewkowycz et al., 2022), and HMMT (Balunović et al., 2025).

**Evaluation Settings.** For evaluation generation, we use a temperature of $0.6$ and top-$p$ of $0.95$ to allow for diverse reasoning paths, with a maximum token limit of 32768 to accommodate long chain-of-thought processes. The random seed is fixed at 42 for reproducibility. Considering that standard benchmarks, such as AIME, contain a relatively small number of questions, the statistical variation of the evaluation results can be significantly influenced. To mitigate this issue and enhance the robustness of our metrics, we compute the $18.0$ $Avg@k$ for each problem, defined as the average accuracy across $k$ generations. We also evaluate Pass@1 in $k$ *i.e.,* if there is on correct answer in $k$ generations, we consider this problem as solved.

## 3. Results and Analysis

### 3.1. Main Results

**LoRA is Definitely not the optimal choice for RL.** A salient observation from our experiments is the consistent superiority of some LoRA-variants *e.g.,* DoRA and MiSS.

> **Finding 1:** Standard LoRA is **suboptimal** for RLVR. **Structural variants**, that decouple learning dynamics (DoRA), sharding parameters (MiSS) or allocate parameters adaptively (AdaLoRA) , currently represent the optimal parameter-efficient choices for RLVR beyond LoRA. **So stop using LoRA for RLVR!**

While standard LoRA ($42.5\%$) serves as a respectable baseline, it consistently trails behind full-parameter fine-tuning

($44.9\%$), suggesting a limitation in its rigid low-rank constraint when facing the complex policy shifts required by RL. In contrast, structural variants effectively bridge or even exceed this gap. DoRA breaks the ceiling with an overall average of $46.6\%$, surpassing the full-parameter baseline across multiple benchmarks (*e.g.,* AIME and AMC). Similarly, AdaLoRA ($44.2\%$) and MiSS ($43.4\%$) consistently outperform standard LoRA. We attribute this superiority to the mitigation of the optimization rigidity inherent in standard LoRA, and we hypothesize that this stems from a fundamental alignment between the architectural inductive biases of these variants and the unique optimization dynamics of RLVR.

**Less is not always more for parameter-efficient RLVR.** While recent findings suggest RLVR is compatible with low-rank updates (Schulman & Lab, 2025), our results identify a critical expressivity floor. We observe that while moderate efficiency gains are sustainable, extreme parameter reduction methods fail to capture the complex policy shifts required for reasoning. As detailed in Table 4, there exists a clear boundary in performance based on the adaptation mechanism. Methods that retain low-rank matrix structures, such as LoRA-FA (which freezes projection matrices **A** and trains only **B**), maintain competitive performance comparable to standard LoRA. This indicates that the RLVR signal, though sparse, is sufficient to drive updates in a semi-frozen low-rank subspace.

> **Finding 2: RLVR demands a minimum threshold of expressivity.** While moderate reduction (*e.g.,* LoRA-FA) is effective, extreme parameter reduction methods (*e.g.,* VeRA, IA$^3$) that rely on vector-only updates lack the necessary plasticity to reorient reasoning circuits.

While RLVR can tolerate moderate parameter reduction, we find it fails under *extreme parameter reduction*. For instance, VeRA—which freezes both low-rank matrices and learns only scaling vectors—drops to $40.7\%$ accuracy, and IA$^3$ suffers a severe degradation to $22.3\%$. These results indicate that RLVR requires a minimum threshold of trainable adapter capacity to succeed. Unlike supervised fine-tuning, the optimization process in RLVR appears to demand higher expressivity in the trainable space; reducing this space to mere scaling vectors (as in VeRA, IA$^3$, or LN-tuning) creates a bottleneck that prevents the model from effectively learning complex reasoning behaviors.

**SVD-based Initialization Misaligns with RL Optimization.** The substantial underperformance of initialization strategies derived from Singular Value Decomposition warrants a mechanistic explanation. As shown in Table 3, PiSSA suffers a catastrophic collapse to near-zero accuracy ($0.2\%$), while MiLoRA ($18.0\%$) significantly trails standard baselines. Recent work by Zhu et al. (2025) re-

*Table 3.* Comparison of accuracy and pass scores. All values are reported in percentages.

| Methods | Avg. | AIME24@32 | | AIME25@32 | | AMC@32 | | HMMT@32 | | MATH500@4 | | Minerva@4 | |
|---|---|---|---|---|---|---|---|---|---|---|---|---|---|
| | | Avg. | Pass | Avg. | Pass | Avg. | Pass | Avg. | Pass | Avg. | Pass | Avg. | Pass |
| ***Baseline*** | | | | | | | | | | | | | |
| Base | 40.5 | 32.4 | 76.7 | 22.2 | 33.3 | 69.0 | 90.0 | 9.9 | 23.3 | 72.0 | 86.8 | 14.2 | 25.0 |
| Full | 44.9 | 34.9 | 56.7 | 23.8 | 46.7 | 68.8 | 92.5 | 13.5 | 40.0 | 74.8 | 88.6 | **17.0** | 26.5 |
| LoRA | 42.5 | 33.2 | 60.0 | 22.9 | 36.7 | 64.4 | 95.0 | 13.3 | 33.3 | 72.1 | 87.4 | 13.5 | 23.5 |
| ***Structural*** | | | | | | | | | | | | | |
| AdaLoRA | 44.2 | 29.4 | 53.3 | 26.7 | 50.0 | 68.9 | 95.0 | 14.1 | 40.0 | 73.8 | 88.0 | 15.5 | 27.5 |
| DoRA | **46.6** | **39.0** | 80.0 | **28.8** | 43.3 | **71.9** | 95.0 | 13.4 | 36.7 | **75.8** | 90.0 | 14.7 | 26.5 |
| MiSS | 43.4 | 27.5 | 50.0 | 23.3 | 26.7 | 68.6 | 95.0 | 15.8 | 33.3 | 72.4 | 90.4 | 16.5 | 30.1 |
| ***Efficient*** | | | | | | | | | | | | | |
| LoRA-FA | 43.0 | 29.2 | 53.3 | 22.7 | 46.7 | 65.0 | 95.0 | 15.1 | 36.7 | 73.5 | 87.2 | 15.6 | 26.8 |
| VeRA | 40.7 | 29.1 | 60.0 | 21.7 | 36.7 | 61.5 | 95.0 | 14.4 | 30.0 | 69.7 | 85.6 | 13.1 | 24.6 |
| ***Initialization*** | | | | | | | | | | | | | |
| LoRA+ | 43.9 | 28.1 | 50.0 | 25.9 | 50.0 | 70.0 | 95.0 | **16.5** | 36.7 | 72.2 | 90.4 | 15.4 | 26.5 |
| rsLoRA | 42.3 | 29.2 | 43.3 | 23.3 | 30.0 | 63.4 | 92.5 | 14.4 | 36.7 | 72.6 | 87.6 | 14.4 | 27.2 |
| MiLoRA | 18.0 | 4.2 | 6.7 | 0.0 | 0.0 | 19.6 | 47.5 | 0.0 | 0.0 | 44.5 | 63.4 | 11.7 | 19.9 |
| PiSSA | 0.2 | 0.0 | 0.0 | 0.0 | 0.0 | 0.0 | 0.0 | 0.0 | 0.0 | 0.6 | 1.0 | 0.1 | 0.4 |
| ***Other PEFTs*** | | | | | | | | | | | | | |
| IA$^3$ | 22.3 | 0.7 | 6.7 | 3.5 | 10.0 | 25.2 | 57.5 | 0.6 | 3.3 | 55.4 | 71.4 | 12.8 | 22.1 |
| LN Tuning | 41.8 | 31.7 | 56.7 | 23.8 | 26.7 | 65.1 | 95.0 | 11.8 | 23.3 | 69.9 | 85.0 | 14.1 | 25.0 |

*Figure 2.* **Left 1:** Normalized magnitude of updates across singular value indices. **Left 2**: Cumulative proportion of energy explained by the top-$k$ components. **Right1:** Accuracy reward curves during training, illustrating the performance collapse of SVD-based initializations in the RLVR setting compared to standard baselines. **Right2:** Accuracy reward of batch size 128 and 32.

*Table 4.* Trainable parameters and overall accuracy of full-parameter fine-tuning, standard LoRA, MiSS, VeRA and LN Tuning.

| Method | Train Param. | Overall Acc. (%) |
|---|---|---|
| Full | 100% | **44.9** |
| LoRA | 1.55% | 42.5 |
| LoRA Rank 1 | 0.0015% | 40.5 |
| MiSS | 0.99% | 43.4 |
| VeRA | 0.0029% | 40.7 |
| LN Tuning | 0.0035% | 41.8 |

veals that RLVR operates in an *off-principal regime*: unlike SFT, which targets high-magnitude principal weights, RLVR updates consistently localize to low-curvature, non-principal subspaces to preserve the pre-trained spectral geometry. Based on this characteristic, PiSSA fails predictably: by explicitly restricting updates to the principal subspace ($U_{[:r]}, V_{[:r]}$), it imposes a structural bias that directly conflicts with the intrinsic requirement of RLVR, leading to the observed collapse (0.2% accuracy).

A more critical finding, however, contradicts the intuitive extension of the off-principal theory. Theoretically, MiLoRA initializes adapters using minor singular components ($U_{[r:]}, V_{[r:]}$), which presumably aligns with the off-principal nature of RLVR. Yet, our empirical results (see Figure 2, Right) show it first trains with a well reward in-

creasing and then fails to converge (18.0%). We analyze this failure through the lens of spectral analysis on weight updates. Our Full Fine-Tuning results (Figure 2, Left, Green line) exhibit a uniform distribution of updates across the entire singular value spectrum, and Figure 2 (Left, Blue line) uncovers the mechanism: despite being initialized in the off-principal subspace, the final updates of MiLoRA exhibit a sharp spike at the dominant principal components ($k \approx 0$), behaving almost identically to PiSSA.

Despite the theoretical alignment, MiLoRA fails due to the disparity between initialization and gradient flow. We formalize the update dynamics at step $t$ as:

$$\Delta \boldsymbol{W}_{t+1} \leftarrow \Delta \boldsymbol{W}_t - \eta \nabla \mathcal{L}(\boldsymbol{W}_t),$$
$$\|\Delta \boldsymbol{W}_0\|_F = \|\boldsymbol{B}_{\text{init}} \boldsymbol{A}_{\text{init}}\|_F \approx 0. \quad (3)$$

The minor singular values used for initialization satisfy $\sigma_{tail} \rightarrow 0$. Consequently, the initial adapter state effectively collapses to zero. This renders the intended structural constraint numerically non-existent. Without a significant initial bias $\|\Delta \boldsymbol{W}_0\|_F$, the optimization trajectory is dictated by the spectral properties of the gradient $\nabla \mathcal{L}$. Since the gradient aligns with the directions of maximum variance (principal components $\boldsymbol{U}_{:k}$), the update projects onto the principal

subspace where $\langle \nabla\mathcal{L}, \boldsymbol{U}_{:k} \rangle \gg \langle \nabla\mathcal{L}, \boldsymbol{U}_{k:} \rangle$. Consequently, the updates are immediately reoriented by the dominant principal gradients, causing the model to degenerate from the off-principal regime back to the principal subspace, as evidenced by the spectral spike in Figure 2.

> **Finding 3:** Existing SVD-based initialization strategies are unsuitable for RLVR. PiSSA fails due to a design conflict (forcing principal updates), creating a structural mismatch with RLVR's intrinsic tendency to **learn off the principals** (Zhu et al., 2025), resulting in training collapse. However, we find that MiLoRA fails due to optimization instability—negligible initialization magnitude causes the model to **degenerate back into principal-component updates**, failing to maintain the necessary off-principal trajectory.

*Table 5.* Hyperparameters for RLVR training across model scales.

| Ablation Term | Settings |
| --- | --- |
| Batch Size | 32, 128 |
| RLVR Algorithm | GRPO, DAPO, Dr. GRPO |
| Learning Rate | $1 \times 10^{-5}, 5 \times 10^{-6}, 1 \times 10^{-6}$ |
| Rank | 1, 8, 16, 32 |

### 3.2. Ablation Studies

To rigorously validate the robustness of our findings and disentangle the influence of hyperparameter choices from intrinsic method efficacy, we conducted a comprehensive series of ablation studies. As summarized in Table 5, we systematically varied key training configurations across four orthogonal dimensions: batch size, reinforcement learning algorithms, learning rates, and LoRA ranks.

**RLVR Training Batch Size.** Recent empirical studies in SFT suggest that LoRA efficacy is inversely correlated with batch size, favoring a *small-batch, high-frequency* update regime (Schulman & Lab, 2025). The prevailing hypothesis attributes this to the high information density of SFT's hard teacher-forcing objective, which can saturate the limited capacity of low-rank adapters when processing large batches. We hypothesized that this constraint would be relaxed in RLVR, as the supervision signal consists only of sparse, scalar rewards rather than dense token-level targets. Our results (see Table 7 and Figure 2) confirm that: reducing the batch size to 32 yielded an slightly higher average accuracy of 42.5%. Notably, on the challenging AIME 2024 benchmark, the larger batch size actually outperformed the smaller one. This indicates that the *small batch* heuristic from SFT does transfer to RLVR but perform less well.

**RLVR Algorithms.** We further investigated whether the efficacy of parameter-efficient methods is sensitive to the specific formulation of the reinforcement learning objective. We evaluated standard LoRA across three representative RLVR algorithms: GRPO (Shao et al., 2024), DAPO (Yu et al., 2025), and Dr. GRPO (Liu et al., 2025), which em-

ploy varying strategies for advantage estimation and regularization. Our experiments reveal a remarkable degree of algorithmic invariance; the performance of LoRA and other PEFT methods remains consistent across these methods, with no statistically significant deviation in reasoning accuracy. This suggests that the effectiveness of parameter efficient methods in this domain is driven by the fundamental dynamics of learning from sparse, verifiable rewards, rather than being contingent on specific loss function nuances (such as the specific implementation of KL penalties or ratio clipping). Consequently, optimal PEFT choices like DoRA are likely transferable across the broader landscape of RLVR algorithms.

**Learning Rate.** Our results (see Table 7) corroborate the scaling laws proposed in Schulman & Lab (2025), confirming that learning rate magnitude is a decisive factor in RLVR stability. We observe that the optimal performance was consistently achieved at scales $\mathrm{LR} = M_{\mathrm{LoRA}} \cdot \left( \frac{2000}{\text{hidden size}} \right)^{\text{model pow}+\text{LoRA pow}}$, validating that careful learning rate scaling is as critical as the choice of the PEFT method itself.

**LoRA Rank.** Regarding the rank dimension, we challenge the notion that minimal ranks are sufficient for maximizing RL performance. While prior works suggest that even Rank=1 adapters can complete RLVR tasks effectively (Mukherjee et al., 2025), our ablation across ranks 1, 8, 16, 32 reveals that relatively high ranks *e.g.,* 16 and 32 yield superior results. Specifically, setting $r = 1$ consistently underperformed higher-rank configurations. Given that the parameter overhead of LoRA is still negligible compared to the base model size, we advocate for avoiding extreme rank reduction; maintaining a moderate rank ensures sufficient expressivity to capture complex reasoning adjustments without compromising computational efficiency.

### 3.3. Scaling on Stronger Models

To verify the generalizability of our findings, we scale our evaluation to the 7B parameter regime using *DeepSeek-R1-Distill-Qwen-7B*. As summarized in Table 6, the relative performance hierarchy observed in smaller models remains largely consistent at this larger scale. Both **DoRA** and **LoRA+** achieve an overall average accuracy of 55.0%, outperforming the standard LoRA baseline (54.8%). This consistent superiority suggests that the advantages of magnitude-direction decoupling (in DoRA) and optimized learning rate ratios (in LoRA+) are not mere artifacts of smaller model scales, but are intrinsic to the RLVR optimization landscape. Notably, DoRA maintains its lead across several challenging benchmarks, such as AMC (83.1%) and AIME25 (38.7%). These results reinforce our conclusion that for large-scale reasoning models, employing architecturally enhanced or optimization-aware awdapters is more effective than relying on the standard LoRA formulation.

Table 6. Comparison of accuracy (Avg.) and pass scores (Pass). All values are reported in percentages (std dev removed for clarity).

| Methods | Avg. | AIME24@32 | | AIME25@32 | | AMC@32 | | HMMT@32 | | MATH500@4 | | Minerva@4 | |
|---|---|---|---|---|---|---|---|---|---|---|---|---|---|
| | | Avg. | Pass | Avg. | Pass | Avg. | Pass | Avg. | Pass | Avg. | Pass | Avg. | Pass |
| **7B Model** | | | | | | | | | | | | | |
| LoRA | 54.8 | **48.3** | 73.3 | 35.9 | 73.3 | 81.4 | 97.5 | 22.0 | 53.3 | 80.9 | 94.8 | 27.2 | 40.1 |
| DoRA | 55.0 | 45.8 | 70.0 | 38.7 | 66.7 | **83.1** | 97.5 | **23.7** | 56.7 | 80.1 | 93.2 | 25.9 | 39.0 |
| MiSS | 53.4 | 42.5 | 63.3 | 36.4 | 70.0 | 77.9 | 97.5 | 22.2 | 60.0 | 80.8 | 93.2 | 26.5 | 39.0 |
| LoRA+ | **55.5** | 46.3 | 73.3 | **39.0** | 70.0 | 82.7 | 100.0 | 22.0 | 66.7 | **81.8** | 94.6 | **27.5** | 40.8 |

Table 7. Comparison of accuracy and pass scores. All values are reported in percentages (std dev removed for clarity).

| Methods | Avg. | AIME24@32 | | AIME25@32 | | AMC@32 | | HMMT@32 | | MATH500@4 | | Minerva@4 | |
|---|---|---|---|---|---|---|---|---|---|---|---|---|---|
| | | Avg. | Pass | Avg. | Pass | Avg. | Pass | Avg. | Pass | Avg. | Pass | Avg. | Pass |
| **Baseline** | | | | | | | | | | | | | |
| Full | **44.9** | **34.9** | 56.7 | **23.8** | 46.7 | **68.8** | 92.5 | **13.5** | 40.0 | **74.8** | 88.6 | **17.0** | 26.5 |
| LoRA | 42.5 | 33.2 | 60.0 | 22.9 | 36.7 | 64.4 | 95.0 | 13.3 | 33.3 | 72.1 | 87.4 | 13.5 | 23.5 |
| - *Bsz* 128, *learning rate* $1 \times 10^{-5}$, *DAPO* | | | | | | | | | | | | | |
| **Batch Size** | | | | | | | | | | | | | |
| 32 Bsz | 43.0 | 28.4 | 50.0 | 24.7 | 43.3 | 67.6 | 90.0 | 15.0 | 33.3 | 72.0 | 86.8 | 14.3 | 25.0 |
| **Learning Rate** | | | | | | | | | | | | | |
| $1 \times 10^{-5}$ | 42.3 | 29.2 | 43.3 | **23.3** | 30.0 | 63.4 | 92.5 | **14.4** | 36.7 | 72.6 | 87.6 | 14.4 | 27.2 |
| $5 \times 10^{-6}$ | 42.3 | **30.4** | 50.0 | 18.2 | 36.7 | **65.8** | 92.5 | 13.9 | 40.0 | **73.1** | 87.2 | **14.9** | 27.9 |
| **Rank** | | | | | | | | | | | | | |
| 1 | 40.5 | 25.7 | 53.3 | 21.7 | 36.7 | 61.7 | 95.0 | 13.4 | 23.3 | 70.6 | 86.4 | 13.9 | 24.3 |
| 8 | 42.3 | 28.9 | 43.3 | 22.1 | 40.0 | 65.4 | 97.5 | 15.4 | 40.0 | 72.2 | 87.0 | 13.5 | 25.4 |
| 16 | **43.9** | **31.2** | 56.7 | **23.8** | 36.7 | **67.3** | 95.0 | 14.7 | 43.3 | **73.8** | 89.6 | **16.1** | 29.4 |
| **Algorithm** | | | | | | | | | | | | | |
| Dr. GRPO | **42.0** | **28.9** | 50.0 | **25.4** | 43.3 | **65.6** | 97.5 | **13.0** | 33.3 | 70.1 | 87.6 | 14.5 | 28.3 |
| GRPO | 40.5 | 25.2 | 43.3 | 15.8 | 33.3 | 64.5 | 97.5 | 12.7 | 30.0 | **71.0** | 86.8 | **15.9** | 28.3 |

## 4. Future Work

**Advanced Infrastructure and Scaling.** While TRL affords broad PEFT compatibility, its limitations in large-scale distributed training motivate a migration to higher-performance frameworks like VeRL. This upgrade will let us expand evaluation beyond the DeepSeek-R1-Distill family and short-horizon schedules, testing *R1-Zero-like* cold-start paradigms and diverse architectures under prolonged training to verify the stability and asymptotic performance of PEFT-RL at scale.

**Mechanistic Interpretability of Adapter Dynamics.** We observe that structural variants (*e.g.,* DoRA) align with RLVR while SVD-based initializations (*e.g.,* PiSSA) collapse, but the underlying mathematics remains unclear. We plan to study the spectral evolution and optimization landscapes of these adapters, moving toward a grounded theory of why specific structural biases suit the sparse, off-principal optimization of RL.

**Broader Frontiers and Deployment Stability.** Finally, we aim to test the universality of our findings across multimodal, multi-turn, and asynchronous RL settings, while addressing often-overlooked deployment challenges such as the numerical stability of weight merging and train-inference inconsistencies—issues essential for moving PEFT-RL from benchmarking to real-world application.

## 5. Conclusion

We present the first systematic, large-scale evaluation of PEFT methods under the RLVR paradigm, spanning 12+ variants across multiple model scales.We present the first systematic, large-scale evaluation of PEFT methods under the RLVR paradigm, spanning 12+ variants across multiple model scales. Several findings challenge the default reliance on standard LoRA. Structural variants consistently outperform LoRA and can even exceed full-parameter fine-tuning, underscoring the value of magnitude-direction decoupling for RL's complex policy shifts. Conversely, SVD-informed initialization methods (PiSSA, MiLoRA) suffer structural misalignment: their focus on principal components conflicts with RLVR's off-principal update dynamics, causing instability or collapse. We also identify an expressivity floor, where extreme parameter reduction (e.g., VeRA, Rank-1 adapters) creates a structural bottleneck that limits reasoning plasticity. Overall, we advocate moving beyond default LoRA toward geometry-aware adapters like DoRA, which better balance efficiency and reasoning capability.

## Impact Statement

This paper presents work whose goal is to advance the field of Machine Learning. There are many potential social consequences of our work, none which we feel must be specifically highlighted here.

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

# A. Implementation Details

## A.1. Supported PEFT Methods

Table 8 summarizes all parameter-efficient methods supported in our framework, along with their corresponding implementation strategies.

*Table 8.* Supported PEFT methods in our framework and their implementation details.

| Method Key | Backend | Notes |
| --- | --- | --- |
| `lora` | `LoraConfig` | Standard LoRA |
| `dora` | `LoraConfig(use_dora=True)` | Weight-decomposed |
| `adalora` | `AdaLoraConfig` | Adaptive rank allocation |
| `pissa` | `LoraConfig(init_lora_weights="pissa_niter_4")` | Principal SVD init |
| `milora` | Custom SVD (mode="min") | Minor components init |
| `milora_plus` | Custom SVD (direction-only) | Off-principal directions |
| `vera` | `VeraConfig` | Vector-based adaptation |
| `miss` | `MissConfig` | Mixture of sub-spaces |
| `lora_plus` | `LoraConfig` + `create_loraplus_optimizer` | Differentiated LR |
| `lorafa` | `LoraConfig` + `create_lorafa_optimizer` | Frozen A matrix |
| `layernorm` | `LNTuningConfig` | LayerNorm only |
| `IA3` | `IA3Config` | Activation scaling |
| `rslora` | `LoraConfig(use_rslora=True)` | Rank-stabilized scaling |

## A.2. MiLoRA Initialization Algorithm

Algorithm 1 presents the SVD-based initialization strategy for MiLoRA, which initializes adapter matrices using the minor singular components of the pre-trained weight matrix.

---

**Algorithm 1** MiLoRA: Minor Singular Components Initialization

---

**Require:** Pre-trained weight matrix $\mathbf{W}_0 \in \mathbb{R}^{d_{\text{out}} \times d_{\text{in}}}$, rank $r$, scaling factor $\alpha$
**Ensure:** Initialized LoRA matrices $\mathbf{A}, \mathbf{B}$ and modified base weights
 1: $\mathbf{U}, \mathbf{S}, \mathbf{V} \leftarrow \text{SVD}(\mathbf{W}_0)$ {Full SVD decomposition}
 2: $\mathbf{U}_{\text{sel}} \leftarrow \mathbf{U}[:, -r:]$ {Select last $r$ columns (minor components)}
 3: $\mathbf{S}_{\text{sel}} \leftarrow \mathbf{S}[-r:]$ {Select smallest $r$ singular values}
 4: $\mathbf{V}_{\text{sel}} \leftarrow \mathbf{V}[-r:,:]$ {Select last $r$ rows}
 5: scaling $\leftarrow \alpha/r$
 6: $\mathbf{S}_{\text{sel}} \leftarrow \mathbf{S}_{\text{sel}}/\text{scaling}$
 7: $\mathbf{S}_{\sqrt{}} \leftarrow \sqrt{\mathbf{S}_{\text{sel}}}$
 8: $\mathbf{B} \leftarrow \mathbf{U}_{\text{sel}} \cdot \text{diag}(\mathbf{S}_{\sqrt{}})$ {$\mathbf{B} \in \mathbb{R}^{d_{\text{out}} \times r}$}
 9: $\mathbf{A} \leftarrow \text{diag}(\mathbf{S}_{\sqrt{}}) \cdot \mathbf{V}_{\text{sel}}$ {$\mathbf{A} \in \mathbb{R}^{r \times d_{\text{in}}}$}
10: $\Delta \leftarrow \text{scaling} \cdot \mathbf{B} \cdot \mathbf{A}$
11: $\mathbf{W}_0 \leftarrow \mathbf{W}_0 - \Delta$ {Subtract from base weights}
12: **return** $\mathbf{A}, \mathbf{B}$

---

## A.3. MiLoRA++ Initialization Algorithm

Algorithm 2 presents MiLoRA++, an improved variant that uses direction-only initialization without modifying the base model weights. This addresses the vanishing gradient issue observed in standard MiLoRA.

**Key differences from MiLoRA:**

- Discards singular values to avoid vanishing gradients from tiny $\sigma_{\text{min}}$

- Initializes $\mathbf{B}$ to zero instead of SVD-derived values

---

**Algorithm 2** MiLoRA++: Direction-Only Off-Principal Initialization

---

**Require:** Pre-trained weight matrix $\mathbf{W}_0 \in \mathbb{R}^{d_{\text{out}} \times d_{\text{in}}}$, rank $r$
**Ensure:** Initialized LoRA matrices $\mathbf{A}, \mathbf{B}$
1: $\mathbf{U}, \mathbf{S}, \mathbf{V}^T \leftarrow \text{SVD}(\mathbf{W}_0)$ $\{\mathbf{V}^T$ is V transpose$\}$
2: $\mathbf{V}_{\text{sel}} \leftarrow \mathbf{V}^T[-r :, :]$ $\{$Off-principal directions (last $r$ rows)$\}$
3: $\mathbf{A} \leftarrow \mathbf{V}_{\text{sel}}$ $\{$Unit-scale orthonormal vectors$\}$
4: $\mathbf{B} \leftarrow \mathbf{0}^{d_{\text{out}} \times r}$ $\{$Zero initialization for stable start$\}$
5: $\{$Do NOT modify base model weights $(\Delta = 0)\}$
6: **return** $\mathbf{A}, \mathbf{B}$

---

- Does not subtract $\Delta$ from base weights, ensuring identity mapping at initialization

- Preserves orthonormal directions for clean gradient flow into off-principal subspace

### A.4. Reward Function

Algorithm 3 describes the mathematical answer verification process used to compute binary rewards.

---

**Algorithm 3** Mathematical Answer Verification for RLVR

---

 1: **function** EXTRACTBOXEDANSWER(text)
 2:    idx $\leftarrow$ rfind(text, `"\boxed"`) $\{$Find last occurrence$\}$
 3: **if** idx $< 0$ **then**
 4:    idx $\leftarrow$ rfind(text, `"\fbox"`)
 5: **end if**
 6: **if** idx $< 0$ **then**
 7:    **return** None
 8: **end if**
 9:    Find matching closing brace via brace counting
10: **return** inner content of `\boxed{...}`
11: **end function**
12:
13: **function** COMPUTESCORE(prediction, ground_truth)
14:    pred_fmt $\leftarrow$ `"$"` + prediction + `"$"`
15:    gold_fmt $\leftarrow$ `"$"` + ground_truth + `"$"`
16:    verify_func $\leftarrow$ math_metric(LatexConfig, ExprConfig)
17:    **try:**
18:      score, _ $\leftarrow$ verify_func([gold_fmt], [pred_fmt])
19:      **return** float(score) $\in \{0, 1\}$
20:    **except** TimeoutException:
21:      **return** 0.0
22: **end function**

---

## B. Experimental Configuration

### B.1. Target Modules

All LoRA-based methods target the following linear projection modules in the transformer architecture:

```
target_modules = [
    "q_proj",    # Query projection
    "k_proj",    # Key projection
    "v_proj",    # Value projection
    "o_proj",    # Output projection
```

```
    "gate_proj", # Gate projection (MLP)
    "up_proj",   # Up projection (MLP)
    "down_proj"  # Down projection (MLP)
]
```

This configuration follows the recommendation from prior work (Schulman & Lab, 2025; Wang et al., 2025b) that targeting all linear modules yields superior performance for RLVR tasks.

### B.2. Training Infrastructure

We utilize DeepSpeed ZeRO Stage 2 for distributed training. The configuration is as follows:

```
compute_environment: LOCAL_MACHINE
deepspeed_config:
  zero_stage: 2
  offload_optimizer_device: none
  offload_param_device: none
  zero3_init_flag: false
  gradient_clipping: auto
distributed_type: DEEPSPEED
mixed_precision: bf16
num_processes: 4
```

### B.3. Software Dependencies

Table 9 lists the core software dependencies used in our experiments.

*Table 9.* Software dependencies for the PeRL framework.

| Category | Packages |
|---|---|
| Deep Learning | `torch, transformers, peft` |
| RL Training | `trl[vllm], accelerate, deepspeed` |
| Data | `datasets, huggingface-hub` |
| Evaluation | `math-verify` |
| Logging | `wandb` |
| Utilities | `fire` |

## C. Evaluation Protocol

### C.1. Evaluation Datasets

Table 10 provides the complete information for all evaluation benchmarks used in our study.

*Table 10.* Evaluation datasets with HuggingFace identifiers and data splits.

| Dataset | HuggingFace ID | Split | Size |
|---|---|---|---|
| AIME 2024 | `HuggingFaceH4/aime_2024` | train | 30 |
| AIME 2025 | `yentinglin/aime_2025` | train | 30 |
| AMC 2023 | `zwhe99/amc23` | test | 40 |
| MATH-500 | `HuggingFaceH4/MATH-500` | test | 500 |
| Minerva | `math-ai/minervamath` | test | 272 |
| HMMT 2025 | `FlagEval/HMMT_2025` | train | 30 |

## C.2. Prompt Templates

We use the following prompt template for evaluation (LightEval format):

> **Evaluation Prompt Template**
>
> ```
> {problem} Please reason step by step, and put your final answer within
> \boxed{}.
> ```

## C.3. System Prompt

The following system prompt is used during training to enforce structured reasoning output:

> **Training System Prompt**
>
> ```
> You are a helpful AI Assistant that provides well-reasoned and detailed
> responses.  You first think about the reasoning process as an internal monologue
> and then provide the user with the answer.  Respond in the following format:
> <think>\n...\n</think>\n, then answer.
> ```

## C.4. Evaluation Metrics

For each benchmark, we compute the following metrics:

- **Avg@$k$**: Average accuracy across $k$ generations per problem
- **Pass@$k$**: Binary indicator whether at least one of $k$ generations is correct

The number of generations $k$ varies by benchmark difficulty:

- AIME24, AIME25, AMC, HMMT: $k = 32$ (competition-level difficulty)
- MATH-500, Minerva: $k = 4$ (standard difficulty)

## C.5. Generation Parameters

During evaluation, we use the following generation parameters:

*Table 11.* Generation parameters for evaluation.

| Parameter | Value |
| --- | --- |
| Temperature | 0.6 |
| Top-$p$ | 0.95 |
| Max tokens | 32768 |
| Random seed | 42 |

# D. Additional Training Details

## D.1. GRPO Objective

For completeness, we provide the full GRPO objective used in our experiments. Given a prompt $q$ and a group of $G$ sampled responses $\{o_1, \ldots, o_G\}$, the objective is:

$$\mathcal{J}_{\text{GRPO}}(\theta) = \mathbb{E}_{q \sim \mathcal{D}, \{o_i\} \sim \pi_{\theta_{\text{old}}}} \left[ \frac{1}{G} \sum_{i=1}^{G} \frac{1}{|o_i|} \sum_{t=1}^{|o_i|} \min \left( r_{i,t}(\theta)\hat{A}_i, \text{clip}(r_{i,t}(\theta), 1 \pm \epsilon)\hat{A}_i \right) \right] \tag{4}$$

where $r_{i,t}(\theta) = \frac{\pi_\theta(o_{i,t}|q,o_{i,<t})}{\pi_{\theta_{\text{old}}}(o_{i,t}|q,o_{i,<t})}$ is the probability ratio, and $\hat{A}_i = \frac{R_i - \text{mean}(\{R_j\})}{\text{std}(\{R_j\})}$ is the standardized advantage.

## D.2. DAPO Modifications

DAPO introduces the following modifications to standard GRPO:

- **Clip-Higher**: Decouples clipping range into $\epsilon_{\text{low}}$ and $\epsilon_{\text{high}}$, with $\epsilon_{\text{high}} = 0.28$ to allow more exploration for low-probability tokens

- **Dynamic Sampling**: Filters prompts where all outputs yield identical rewards (all 0 or all 1)

- **No KL Penalty**: Sets $\beta = 0$ for the KL divergence term

## D.3. Training Configuration Details

*Table 12.* Common training configuration for all experiments.

| Parameter | Value |
|---|---|
| Number of generations per prompt ($G$) | 8 |
| Max prompt length | 512 |
| Max completion length | 16384 |
| Gradient accumulation steps | 8 |
| Learning rate scheduler | Constant |
| Warmup ratio | 0.0 |
| KL coefficient ($\beta$) | 0.0 |
| Optimizer | AdamW |
| Precision | BFloat16 |
| vLLM mode | Colocate |
| vLLM GPU memory utilization | 0.4 |

# E. Extended Experiments and Robustness Analysis

This appendix consolidates a comprehensive set of additional experiments that probe the robustness, generality, and limitations of our main findings. We organize the analysis along six axes: (i) an evaluation-protocol correction and re-evaluation against official baselines (**??**); (ii) per-method learning-rate sensitivity (**??**); (iii) rank robustness (**??**); (iv) generalization across domains and model families (**??**); (v) generalization across RLVR algorithms (**??**); and (vi) mechanistic, scaling, and cost analyses (Sections E.1 and E.3 and **??**). Together these results indicate that our three central conclusions—structural-variant superiority, SVD-initialization collapse, and an expressivity floor under extreme parameter reduction—hold across hyperparameters, ranks, domains, model families, and RL algorithms.

## E.1. Spectral Analysis of Weight Updates

Our mechanistic account holds that RLVR favors *off-principal* weight updates, so methods that force updates into the principal subspace (PiSSA, MiLoRA) collapse. We quantify this by measuring the fraction of update energy captured by the top principal components.

## E.2. Robustness to Answer-Formatting Prompts

To verify that the evaluation bug (**??**) affected all methods uniformly, we tested four answer-formatting instructions and measured overall accuracy (Table 13). All methods shift by a similar margin between the corrected \boxed{} prompt and the original (no-instruction) setting, and the relative ordering is preserved across all four prompt formats. The bug therefore does not differentially advantage any method.

*Table 13.* Overall accuracy (%) under four answer-formatting prompts on 1.5B. Rankings are preserved across formats.

| Prompt Format | DoRA | LoRA | LoRA+ | MiLoRA | PiSSA |
|---|---|---|---|---|---|
| `\boxed{}` (corrected) | 56.4 | 52.5 | 55.6 | 23.9 | 0.3 |
| No instruction (original) | 46.6 | 42.5 | 43.9 | 18.0 | 0.2 |
| `### Response:` | 45.3 | 41.2 | 42.6 | 17.3 | 0.1 |
| `ANSWER:` | 44.9 | 40.8 | 42.2 | 16.8 | 0.2 |

*Table 14.* Training cost on 1.5B (4×H200). Rollout dominates wall-clock for all methods.

| Method | Params (%) | Wall-clock (h/100 steps) | Peak GPU Mem (GB) |
|---|---|---|---|
| Full FT | 100 | 4.2 | 78.3 |
| AdaLoRA | 1.62 | 4.0 | 55.6 |
| DoRA | 1.58 | 3.9 | 54.3 |
| LoRA | 1.55 | 3.8 | 52.1 |
| LoRA-FA | 0.78 | 3.8 | 48.5 |
| VeRA | 0.003 | 3.9 | 45.2 |

## E.3. Computational Cost Analysis

Table 14 reports trainable-parameter percentage, wall-clock time, and peak GPU memory on 1.5B (4×H200). Across PEFT methods the wall-clock differences are small because rollout dominates training time (>90% of wall-clock for all methods regardless of PEFT type); the main differentiator is peak memory, where full fine-tuning is substantially heavier. DoRA's accuracy gains thus come at negligible additional training cost relative to LoRA.

